# Trends in Early Sexual Initiation and Its Association with Socio-Environmental Factors among Korean Adolescents

**DOI:** 10.3390/children10040613

**Published:** 2023-03-24

**Authors:** Dong-Hee Ryu

**Affiliations:** Department of Preventive Medicine, Daegu Catholic University School of Medicine, Daegu 42472, Republic of Korea; ryudh@cu.ac.kr; Tel.: +82-53-650-4494

**Keywords:** adolescent, age of onset, coitus, sexual behavior

## Abstract

There has been no research on the trends in early sexual initiation associated with socio-environmental factors in Korea. This study aimed to examine the trends of early sexual initiation with various socio-environmental factors among adolescents. The Korea Youth Risk Behavior Web-based Survey data were used, and two pooled datasets extracted from the 2006–2008 and 2014–2016 waves were conducted and compared. In this study, early sexual initiation was defined as sexual intercourse at the age of 13 years or younger. The weighted percentage and 95% confidence interval reporting early sexual initiation were estimated, and multiple logistic regression was performed for each subgroup of socio-environmental variables with the 2006–2008 pooled data as a reference. Regardless of sex, the weighted percentage of adolescents with sexual experience who reported early sexual initiation increased in 2014–2016 with statistical significance in most cases. Moreover, the possibility of experiencing early sexual intercourse overtime was more prominent among girls than boys. While indifference to adolescent sexual behavior continues, more adolescents become involved in early sexual initiation. Socio-environmental considerations, such as the establishment of safe environments for adolescents’ sexual activity, along with systematized monitoring mechanisms, should be administered.

## 1. Introduction

Adolescence is a special period in which people are involved in active social activities in schools and communities, while immature adolescents are in a state of physical development and are dependent on their parents economically. Adolescence is the time when the first sexual experience occurs [1,2]. The timing of sexual debut is important due to the following reasons. Adolescents can experience disrupted and distorted physical, mental, and social functioning when they initiate sexual activity without having necessary knowledge [3]. In addition, the timing of sexual debut is influenced by personal and social circumstances [4].

With growing concern about early sexual intercourse among adolescents since the 1990s [5], various studies have been conducted in Western countries in this regard. These studies can be classified into those that focus on the health outcomes of early sexual intercourse, and those that have explored determinants associated with early sexual intercourse.

Health outcomes following sexual activity in adolescence include an increased risk of sexually transmitted infections, the psychosocial influence due to unwanted sexual relationships, and other negative health behaviors such as alcohol consumption or substance abuse [6,7]. Most importantly, unintended teenage pregnancy due to non-contraceptive or improper contraceptive measures may be the outcome of heterosexual sexual intercourse that adolescents want to avoid. Pregnancy is a complex issue affecting teens’ physical and mental health. The disruption of everyday life, economic problems, and unstable marital status affect the rest of their lives and those of their families, peers, and educators [8,9]. In addition, unintended teenage pregnancy is associated with preterm delivery, and there is an immediate effect on newborns, such as an increased risk of low birth weight and neonatal death [10,11,12]. Therefore, an examination of the age of first sexual intercourse with the opposite sex, which could lead to unintended teenage pregnancy, should be considered [13].

Studies exploring the determinants of early sexual initiation in adolescents have mostly focused on considering the associated risk factors. The determinants of early sexual experience can be categorized into individual and socio-environmental factors. Most studies focus on socio-environmental factors, such as parenting and education [14,15]. A recent study conducted in Canada reported that a disrupted family structure and low family support were strongly associated with early sexual initiation [14]. Moreover, a study conducted in Korea showed that low household income, low father’s education level, and not living with the family were associated with early sexual initiation [15]. Some studies have targeted the protective factors for early sexual initiation [5,16,17]. One of these studies, which was performed in the United States, found that living with parents, having a high socioeconomic level, living in rural areas, satisfactory academic achievement, and faithfulness were protective factors against postponing sexual activity during adolescence [5]. However, the trends in early sexual initiation associated with socio-environmental factors were rarely studied [18]. A study conducted in the United States in 2019 revealed that rates of early sexual initiation varied by race/ethnicity, location, and maternal educational level [18]. Yet, such research has not yet been conducted in Korea.

In Korea, studies on sexual activity have not attracted attention because of their conservative cultural backgrounds [19,20]. Specifically, research on adolescent sexual activity trends is scarce [21,22]. In this study, it was hypothesized that more Korean adolescents would experience early sexual intercourse over time as in other countries, where an incremental trend is also observed [23]. The study aimed to examine the characteristics of sexual experience with the opposite sex among Korean adolescents and to confirm whether there is a notable change in the prevalence of early sexual initiation. By exploring the trends in various socio-environmental factors, the study intends to identify high-risk groups that require health interventions.

## 2. Materials and Methods

### 2.1. Study Design

It is a cross-sectional study analyzing the Korea Youth Risk Behavior Web-based Survey (KYRBS), an anonymous, self-administered, and computer-aided survey conducted annually by the Korea Disease Control and Prevention Agency (formerly known as the Korea Centers for Disease Control and Prevention) and the Ministry of Education. Stratified multistage cluster sampling was used to obtain a nationally representative sample. A total of 800 sample schools (400 junior high schools and 400 high schools) were allocated since the educational curricula for junior high schools in Korea include 7th to 9th grades, and those for high schools are provided from 10th to 12th grades. A class from each grade was randomly selected, and all students from the selected class except long-term absentees and students with language disorders were included in the survey. Additional details of the KYRBS are provided elsewhere [24]. For the analysis, data were extracted from the 2006–2008 and 2014–2016 waves—hence, two pooled datasets were used. The number of questions vary every survey year. A total of 104, 129, and 131 questions were asked in 2006, 2007, and 2008, respectively. The number of questions asked in 2014, 2015, and 2016 were 125, 125, and 117, respectively. Except for 2016, the number of questions associated with sexual behaviors was ten and that of 2016 decreased to nine.

### 2.2. Participants

From 2006 to 2008, 221,340 students completed the surveys, and 10,255 students (4.6%) reported sexual intercourse with the opposite sex. The total number of students who reported sexual intercourse at the time the participants completed the surveys (2014–2016) was 8682, and the total number of survey participants in this period was 205,631 (4.2%).

Respondents younger than 13 years of age were excluded from the analysis. It was decided that these individuals could have distinguished socio-environmental characteristics that required additional analysis (Appendix A). A total of 336 students (205 boys and 131 girls) from 2006–2008, and 574 students (371 boys and 203 girls) from 2014–2016 pooled data were excluded. All excluded students were 12 years old; hence, students aged 13 to 18 years were included in the analyses. Students who had reported sexual experience but not at the age of sexual initiation were also excluded from the analysis. In addition, some cases that missed reporting school type, in which sex and school sexual composition did not match, and in which age and school type attending could not be matched at a common-sense level were excluded. The total number of analyzed respondents from the 2006–2008 pooled data was 8781 (6148 boys and 2633 girls), and that of the 2014–2016 pooled data was 7176 (5202 boys and 1974 girls). (Figure 1).

### 2.3. Measurements

#### 2.3.1. Age at Sexual Initiation

The KYRBS asked participants whether they had sexual intercourse with the opposite sex or with the same sex. The analysis of this study included people who reported both heterosexual sexual intercourse and the age at first sexual intercourse. There seemed to be no consensus regarding the timing of early sexual initiation since it may differ according to various factors such as time, culture, and historical backgrounds of the society. A study conducted in 2003 defined early sexual initiation as 14 and younger [25]. In this study, early sexual initiation was defined as experiencing sexual intercourse at 13 years and younger, as currently defined in a previous study [18].

#### 2.3.2. Socio-Environmental Factors

Considering the variables provided by the KYRBS [24], the socio-environmental factors are sub-classified into three categories: family, school, and community characteristics. Family characteristics included the level of household income (highest, high, moderate, low, or lowest), residence type (with family, with relatives, with friends/alone, or care facility), and level of parental education (≥college, completed high school, ≤junior high school, or unknown/deceased). Respondents whose residence type was “living with friends/alone” were defined as those living in a school dormitory or boarding house. School characteristics included academic performance level (good–excellent, fair, poor–unsatisfactory), school composition (coed, boys-only, or girls-only), and high school type (general or specialized). More specifically, the school type is classified into junior high school, general high school, and specialized high school. Specialized high schools focus on cultivating talent in specific fields, providing practice-oriented education designed for students who wish to get a job after graduation rather than apply for college. Lastly, community characteristics included the type of community (urban or rural), city type (large cities or provincial areas), and whether the living area was the Seoul Capital Area. Large cities include metropolitan cities (Seoul, Busan, Daegu, Incheon, Daejeon, Gwangju, and Ulsan) and a Special Self-Governing City (Sejong). In contrast, provincial areas include Gangwon, Gyeonggi, Chungbuk, Chungnam, Jeonbuk, Jeonnam, Gyeongbuk, Gyeongnam, and a Special Self-Governing Province (Jeju) according to administrative municipalities. The specialized area in and around the capital city (Seoul) is called the Seoul Capital Area, and includes Seoul, Incheon, and Gyeonggi. About half of Korea’s citizens reside in the Seoul Capital Area, the country’s most modernized and developed region.

### 2.4. Statical Analysis

The weighted percentage and 95% confidence interval (CI) of students reporting sexual onset at 13 years or younger were estimated. The estimation was performed separately according to sex. Chi-squared tests were used for each subgroup of socio-environmental variables to examine statistical differences between the 2006–2008 and 2014–2016 pooled data. The association between early sexual onset and various socio-environmental characteristics was explored using multiple logistic regressions. The reference variables were the 2006–2008 variables, and each model was adjusted for grade and survey year. All statistical analyses were completed with SPSS version 28.0 (BMI, Armonk, NY, USA), and *p*-values of <0.05 were considered to indicate statistical significance. All *p*-values were calculated using two-tailed Wald tests.

## 3. Results

Table 1 presents the socio-environmental characteristics of the students reporting sexual intercourse by sex. Regardless of sex and survey year, most students reported their household income level as moderate, lived with their families, academic performance was poor or unsatisfactory, went to coeducational schools, and that they resided in urban areas. Among boys, it was most common for their parents to have completed high school education in 2006–2008, but this changed to college or above in 2014–2016. Regardless of the survey year, the weighted percentage of boys living in the non-Seoul Capital Area was slightly higher than that of boys living in the Seoul Capital Area, but the opposite was true for girls.

Among male students who had reported sexual experience, 15.2% were involved in early sexual initiation in the 2006–2008 pooled data, and it increased by 3.8% p in the 2014–2016 pooled data (Table 2). Compared to 2006–2008, the weighted percentages of boys reporting early sexual debut had increased in 2014–2016 according to each subgroup of most socio-environmental variables, and statistical significance was shown in some cases. The increased weighted percentages were not exhibited for individuals with the lowest household income, living with friends/alone, whose parental education levels were unknown, or whose parents had already deceased. However, there were no statistically significant differences between the datasets (*p* = 0.377, 0.134, 0.134, and 0.311, respectively).

The weighted percentages of girls reporting early sexual initiation according to their socio-environmental characteristics are shown in Table 3. When an individual’s residence type was a care facility, the weighted percentage decreased in 2014–2016 compared with 2006–2008, but there was no statistically significant difference. Except for the abovementioned factor, the weighted percentage of early sexual onset according to family, school, and community characteristics increased in the 2014–2016 pooled data compared to the 2006–2008 pooled data. Statistical significance was also confirmed in most cases. There were no statistically significant differences for individuals whose household income was the highest (*p* = 0.234), whose parental education levels were lower than junior high school (*p* = 0.151 and 0.941) or whose maternal education level was college or above (*p* = 0.286), and who were living in rural areas (*p* = 0.186).

Table 4 illustrates the results of the multiple logistic regressions adjusted for grade and survey year, indicating an association between early sexual initiation and various socio-environmental characteristics. For both boys and girls, the possibility of experiencing early sexual intercourse was higher with time in most cases. For boys whose household income was the lowest and whose parental education levels were unknown or deceased, the odds ratios (OR) of experiencing early sexual onset were less than one, but there were no statistically significant differences. ORs were less than one without statistical significance for girls living in care facilities (OR: 0.99; 95% CI: 0.13–7.77) and whose maternal education was college or above (OR: 0.86; 95% CI: 0.39–1.91). The possibility of experiencing early sexual intercourse over time was more prominent among girls than boys.

## 4. Discussion

In this study, the 2006–2008 and 2014–2016 pooled data of the KYRBS data were used to examine the trends of early sexual initiation among Korean adolescents. Compared to 2006–2008, it was confirmed that the proportion of adolescents with sexual experience who reported early sexual initiation increased in 2014–2016 with statistical significance. There are studies that have shown that young people tend to have more liberal attitudes regarding the age of first sexual initiation [4,26]. Such an incremental trend toward earlier sexual experience is also observed in other countries in Asia, Latin America, and Australia [23] but has not been observed in European countries in the twenty-first century. According to a study conducted in 2019, there is a decline in age of first sexual initiation among adolescents in Europe [23]. The situation in Europe is different compared to other continents since the use of condom increased in many European countries [27] and most countries have established a legal age of consent, which is the age below which it is illegal to have sex [23].

Another important issue in this study is that compared with the 2006–2008 pooled data, more female students reported early sexual initiation in 2014–2016, and it was more prominent in girls than in boys. Multiple logistic regression results suggest that girls may be more sensitive to school and community environments. This could be interpreted as the result of open-mindedness regarding early sexual experiences among female students. In the past, virginity was commonly defined as abstaining from sexual life until marriage and was a concept mainly applied to females [28]. It is based on sexual norms and gender roles generally accepted in Korean society [17,18,29,30]. It can be assumed that women’s right to self-determine their sexual activity had increased compared to the past when the male initiative triggered the experience. A study in Canada revealed that poor body image, low socioeconomic status, and high social media use influenced early sexual initiation in girls [14]. It was also reported that not living with family, lower socioeconomic status, and early initiation of smoking and alcohol consumption were predictors of early sexual debut among girls [31]. Another study showed that female students were less responsible for safe sexual behavior than males [28]. However, caution should be taken when interpreting the present result because the percentage remains still higher for boys (19.0%) than for girls (16.6%). In addition, boys tend to be more prone to risky health behavior than girls in general [32,33].

Students who live in care facilities and those who live with their relatives showed a high weighted percentage of reporting early sexual initiation in the 2014–2016 pooled data. This suggests that the stability of the family structure is also related to early sexual experience. It can be assumed that a certain level of early sexual initiation is already prevalent among students living in care facilities and that they are more vulnerable to unsafe sexual activity. This result implies the importance of stable family structural characteristics on the timing of sexual debut. The study result correlates with a previous study which figured that black female adolescents living with fathers tended to wait until marriage to have sex [34] and another study which reported that the presence of residential mothers and grandmothers is related to the age of sexual onset [35].

This study results had several social implications. First, it suggests the necessity of mandatory sex education that can transmit knowledge about healthy sexual lives. In the United States, health education is mandatory in most states and formally includes sex education in the curriculum [36]. However, the educational situation in Korea is different in that sex education is not included in the regular curriculum, and intermittent education is provided once a year by a lecturer or a health teacher, who is a licensed nurse positioned at school for care injured or ill students primarily [21]. It is difficult to produce effective results with this kind of education; it only gives the nuance that schools provide some kind of education about sexual behaviors. One of the reasons that schools cannot put sex education into the regular curriculum is that parents do not have open-minded attitudes about sex due to the cultural and social characteristics of the country [17,18,29,30]. In Germany and the Netherlands, systematic and practical sex education is provided based on public policies [37]. Considering the cases of other countries and the results of this study, it is necessary to prepare the basis for the systematization of mandatory sex education in Korea. Rather than conveying messages to avoid sexual activity until marriage, sex education should deliver knowledge for safe sexual activity and deal with the overall sexuality concept, including reproductive health care and sexual orientation. Considering the present findings, the implementation of sex-specific sex education should also be considered. Second, the monitoring system for adolescent sexual behavior should be modified. The KYRBS is the only survey that includes items related to the sexual behavior of adolescents. Moreover, the number of questions associated with sexual behavior in the KYRBS continues to decrease, and the content is simplified. In 2006, there were 10 questions, and sexual experiences were specifically classified into kissing and caressing, heterosexual sexual intercourse, homosexual sexual activity, sexual harassment, and so on [24]. In 2021, the number of questions decreased to seven, and it checked whether the respondent had experienced sexual intercourse regardless of the characteristics of sexual partners or sexual behaviors [38]. In Germany, a nationally representative survey called “Youth Sexuality” was designed in 1980 and was conducted under the supervision of the Federal Center for Health Education (BZgA) [39]. In the United States, sexual behavior surveys are conducted for teenagers and young adults [18]. Similar to the instances mentioned above, the establishment of a cohort-type survey that includes adolescents and young adults should be considered by the associated bureaus.

This study had several limitations. First, the fact that students self-reported their age of sexual initiation may not be fully reliable. Second, it is difficult to determine a causal relationship between early sexual initiation and various socio-environmental factors. Third, students under 13 years of age and those outside school could not be considered. Further research should be conducted, since these adolescents are more likely to show different characteristics than the students analyzed in this study. Other study limitations originate from the KYRBS data characteristics. Due to the KYRBS data limitations regarding sexual behavior such as sexual orientation, its association with socio-environmental variables has not been broadly considered. In addition, the KYRBS does not collect specific personal information such as religion [40] that may influence the timing of sexual debut, so its associations with early sexual initiation were not fully covered in this study. Finally, gender or the role of gender [33] could not be considered as an independent variable in association with early sexual initiation. Despite these limitations, the strength of this study lies in the fact that it is the first to examine trends in early sexual initiation, considering the socio-environmental aspects of adolescents in Korea.

## 5. Conclusions

While indifference to adolescent sexual behavior continues, more adolescents have become involved in early sexual initiation. However, it is doubtful whether these incremental trends have been formed based on sufficient knowledge of healthy sexual behavior. Socio-environmental considerations regarding early sexual initiation trends should be administered, focusing on the physical and social development of adolescents based on a healthy and safe sexual life.

## Figures and Tables

**Figure 1 children-10-00613-f001:**
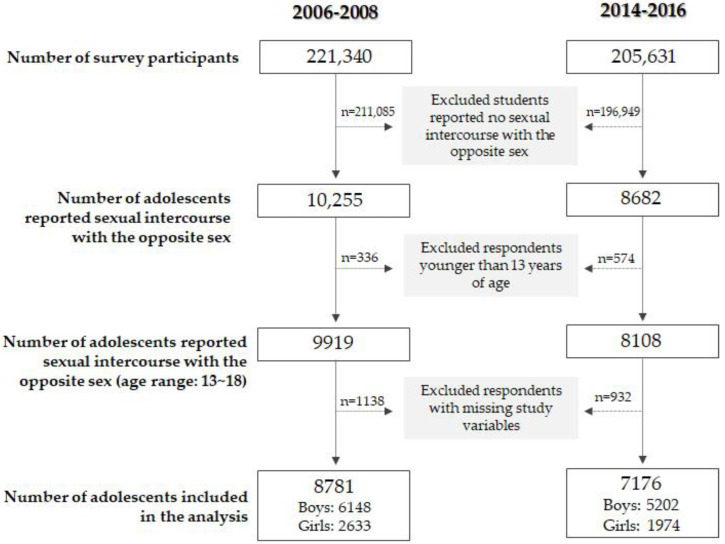
Selection process of study subjects included in the analysis.

**Table 1 children-10-00613-t001:** Socio-environmental characteristics of students reporting sexual intercourse by sex.

Variables	Subgroups	Number (Weighted Percentage)			
		Boys		Girls	
		2006–2008	2014–2016	2006–2008	2014–2016
**Total**		6148 (100.0)	5202 (100.0)	2633 (100.0)	1974 (100.0)
**Family characteristics**					
Household income	Highest	617 (10.4)	776 (15.3)	107 (4.5)	157 (8.2)
	High	1198 (20.3)	1160 (22.5)	400 (15.7)	421 (21.7)
	Moderate	2333 (37.6)	2052 (38.7)	981 (37.9)	763 (37.9)
	Low	1262 (19.8)	823 (16.0)	764 (27.4)	452 (22.6)
	Lowest	738 (11.9)	391 (7.4)	381 (14.6)	181 (9.6)
Residence type	With family	5470 (89.8)	4769 (92.0)	2388 (91.3)	1826 (92.5)
	With relatives	201 (3.0)	91 (1.8)	77 (3.2)	41 (2.3)
	With friends/alone ^1^	337 (4.7)	277 (5.0)	132 (4.2)	79 (3.7)
	Care facility	140 (2.6)	65 (1.3)	36 (1.3)	28 (1.5)
Paternal education	≥College	1998 (34.2)	2353 (46.8)	657 (27.5)	712 (37.1)
	Completed high school	2434 (38.8)	1635 (30.4)	1134 (43.1)	717 (36.2)
	≤Junior high school	664 (9.9)	194 (3.7)	401 (13.3)	93 (4.6)
	Unknown/deceased	1052 (17.1)	1020 (19.1)	441 (16.1)	452 (22.0)
Maternal education	≥College	1336 (23.3)	2003 (39.9)	421 (18.3)	643 (32.9)
	Completed high school	2995 (48.3)	1975 (37.1)	1392 (52.0)	866 (43.8)
	≤Junior high school	742 (11.0)	154 (3.0)	411 (14.3)	96 (5.2)
	Unknown/deceased	1075 (17.3)	1070 (20.0)	409 (15.4)	369 (18.1)
**School characteristics**					
Academic performance	Good–excellent	1772 (29.5)	1687 (32.0)	726 (28.4)	579 (29.8)
	Fair	1399 (22.5)	1213 (22.8)	623 (23.5)	474 (23.7)
	Poor–unsatisfactory	2977 (48.0)	2302 (45.2)	1284 (48.1)	921 (46.5)
Composition	Coed	4007 (65.6)	3358 (62.6)	1648 (66.0)	1399 (68.6)
	Boys only	2141 (34.4)	1844 (37.4)	-	-
	Girls only	-		985 (34.0)	575 (31.4)
School type (high school only)	General	3188 (58.9)	3083 (71.9)	1072 (50.0)	1105 (69.8)
	Specialized ^2^	2223 (41.1)	1229 (28.1)	1173 (50.0)	453 (30.2)
**Community characteristics**					
Community type	Urban (Dong)	5313 (93.4)	4758 (94.3)	2222 (91.8)	1805 (92.5)
	Rural (Eup and Myeon)	835 (6.6)	444 (5.7)	411 (8.2)	169 (7.5)
City type	Large cities ^3^	3016 (47.9)	2242 (43.1)	1165 (44.1)	858 (42.7)
	Provincial areas ^4^	3132 (52.1)	2960 (56.9)	1468 (55.9)	1116 (57.3)
Seoul Capital Area ^5^	Yes	1963 (48.2)	2131 (47.3)	837 (50.8)	878 (50.4)
	No	4185 (51.8)	3071 (52.7)	1796 (49.2)	1096 (49.6)

^1^ With friends/alone: living in a school dormitory or boarding house. ^2^ Specialized high school: high schools that focus on cultivating talent in specific fields, providing practice-oriented education designed for students who wish to get a job after graduation rather than apply for college. ^3^ Large cities: metropolitan cities (Seoul, Busan, Daegu, Incheon, Daejeon, Gwangju, and Ulsan) and the Special Self-Governing City, Sejong. ^4^ Provincial areas: provinces (Gangwon, Gyeonggi, Chungbuk, Chungnam, Jeonbuk, Jeonnam, Gyeongbuk, and Gyeongnam) and the Special Self-Governing Province, Jeju. ^5^ Seoul Capital Area: areas in and around the capital city (Seoul), including Incheon and Gyeonggi.

**Table 2 children-10-00613-t002:** Weighted percentage of boys reporting early sexual initiation.

Variables	Subgroups	Weighted Percentage (95%CI)		*p*-Value *
		2006–2008	2014–2016	
**Total**		15.2 (13.9–16.5)	19.0 (17.7–20.2)	<0.001
**Family characteristics**				
Household income	Highest	33.6 (28.5–38.6)	34.2 (30.4–37.9)	0.862
	High	16.4 (13.5–19.2)	17.6 (15.3–19.9)	0.522
	Moderate	11.6 (10.0–13.2)	16.5 (14.8–18.2)	<0.001
	Low	9.0 (7.2–10.8)	13.6 (11.0–16.2)	0.004
	Lowest	18.6 (14.4–22.8)	16.0 (12.2–19.8)	0.377
	*p*-value	<0.001	<0.001	
Residence type	With family	12.5 (11.2–13.8)	18.0 (16.7–19.3)	<0.001
	With relatives	45.5 (37.4–53.7)	45.5 (34.8–56.1)	0.990
	With friends/alone ^1^	23.3 (17.3–29.4)	17.3 (12.2–22.4)	0.134
	Care facility	58.1 (46.2–69.9)	59.5 (46.3–72.8)	0.870
	*p*-value	<0.001	<0.001	
Paternal education	≥College	15.9 (13.8–18.0)	20.5 (18.7–22.4)	0.002
	Completed high school	9.1 (7.5–10.8)	13.1 (11.3–14.8)	0.002
	≤ Junior high school	12.8 (9.2–16.3)	15.1 (9.7–20.6)	0.468
	Unknown/deceased	28.9 (25.3–32.5)	25.2 (22.2–28.2)	0.134
	*p*-value	<0.001	<0.001	
Maternal education	≥College	19.6 (16.8–22.4)	21.0 (18.9–23.0)	0.450
	Completed high school	9.2 (7.8–10.7)	13.2 (11.6–14.9)	<0.001
	≤ Junior high school	11.0 (7.9–14.2)	15.3 (9.2–21.4)	0.194
	Unknown/deceased	28.5 (24.9–32.1)	26.0 (23.1–29.0)	0.311
	*p*-value	<0.001	<0.001	
**School characteristics**				
Academic performance	Good–excellent	23.9 (21.1–26.7)	26.5 (24.0–28.9)	0.191
	Fair	13.4 (11.2–15.6)	17.3 (15.0–19.6)	0.019
	Poor–unsatisfactory	10.7 (9.2–12.2)	14.4 (12.8–16.1)	0.001
	*p*-value	<0.001	<0.001	
Composition	Coed	14.7 (13.1–16.4)	18.7 (17.1–20.4)	0.001
	Boys only	16.1 (13.7–18.4)	19.3 (17.0–21.6)	0.054
	*p*-value	0.372	0.706	
School type (high school only)	General	11.4 (9.9–12.9)	14.4 (12.9–15.8)	0.006
	Specialized ^2^	7.2 (5.8–8.6)	7.4 (5.8–9.1)	0.819
	*p*-value	<0.001	<0.001	
**Community characteristics**				
Community type	Urban (Dong)	15.4 (13.9–16.8)	19.2 (17.9–20.6)	<0.001
	Rural (Eup and Myeon)	12.7 (9.7–15.8)	14.5 (10.1–19.0)	0.479
	*p*-value	0.147	0.074	
City type	Large cities ^3^	14.9 (13.3–16.6)	20.7 (18.8–22.5)	<0.001
	Provincial areas ^4^	15.4 (13.4–17.5)	17.7 (15.9–19.4)	0.117
	*p*-value	0.737	0.023	
Seoul Capital Area ^5^	Yes	14.7 (12.5–16.9)	18.6 (16.8–20.3)	0.003
	No	15.6 (14.1–17.2)	19.4 (17.4–21.4)	0.020
	*p*-value	0.493	0.526	

Chi-squared tests were used to examine statistical difference. * *p*-values: *p*-values for between dataset differences (2006–2008 and 2014–2016); *p*-values of <0.05 indicate statistical significance. ^1^ With friends/alone: living in a school dormitory or boarding house. ^2^ Specialized high school: high schools that focus on cultivating talent in specific fields, providing practice-oriented education designed for students who wish to get a job after graduation rather than apply for college. ^3^ Large cities: metropolitan cities (Seoul, Busan, Daegu, Incheon, Daejeon, Gwangju, and Ulsan) and the Special Self-Governing City, Sejong. ^4^ Provincial areas: provinces (Gangwon, Gyeonggi, Chungbuk, Chungnam, Jeonbuk, Jeonnam, Gyeongbuk, and Gyeongnam) and the Special Self-Governing Province, Jeju. ^5^ Seoul Capital Area: areas in and around the capital city (Seoul), including Incheon and Gyeonggi.

**Table 3 children-10-00613-t003:** Weighted percentage of girls reporting early sexual initiation.

Variables	Subgroups	Weighted Percentage (95%CI)		*p*-Value *
		2006–2008	2014–2016	
**Total**		9.2 (7.6–10.7)	16.6 (14.6–18.6)	<0.001
**Family characteristics**				
Household income	Highest	33.8 (21.8–45.9)	43.1 (34.1–52.0)	0.234
	High	12.0 (8.3–15.7)	19.4 (15.1–23.7)	0.012
	Moderate	8.5 (6.1–10.8)	14.3 (11.6–17.0)	0.002
	Low	4.4 (2.7–6.1)	8.4 (5.6–11.2)	0.011
	Lowest	9.3 (5.5–13.1)	15.4 (9.4–21.4)	0.080
	*p*-value	<0.001	<0.001	
Residence type	With family	8.2 (6.6–9.7)	14.7 (12.7–16.6)	<0.001
	With relatives	17.3 (6.6–27.9)	46.7 (28.7–64.6)	0.004
	With friends/alone ^1^	12.3 (2.7–21.8)	31.7 (19.6–43.8)	0.017
	Care facility	50.5 (29.2–71.8)	49.2 (28.7–69.7)	0.932
	*p*-value	<0.001	<0.001	
Paternal education	≥College	11.7 (8.6–14.8)	18.0 (14.8–21.2)	0.008
	Completed high school	5.7 (4.1–7.3)	10.3 (7.9–12.7)	0.001
	≤ Junior high school	5.8 (3.2–8.5)	10.3 (3.8–16.9)	0.151
	Unknown/deceased	16.9 (12.0–21.8)	25.7 (21.0–30.4)	0.016
	*p*-value	<0.001	<0.001	
Maternal education	≥College	14.4 (10.2–18.6)	17.4 (14.1–20.8)	0.286
	Completed high school	5.6 (4.2–7.0)	10.8 (8.5–13.0)	<0.001
	≤Junior high school	5.5 (2.9–8.2)	5.7 (1.1–10.4)	0.941
	Unknown/deceased	18.4 (13.2–23.5)	32.1 (26.7–37.5)	<0.001
	*p*-value	<0.001	<0.001	
**School characteristics**				
Academic performance	Good–excellent	12.9 (9.7–16.0)	20.5 (16.6–24.4)	0.003
	Fair	8.6 (5.7–11.4)	17.0 (13.2–20.8)	<0.001
	Poor–unsatisfactory	7.3 (5.5–9.1)	13.8 (11.3–16.3)	<0.001
	*p*-value	0.003	0.008	
Composition	Coed	10.8 (8.5–13.0)	19.9 (17.2–22.5)	<0.001
	Girls only	6.1 (4.3–7.9)	9.3 (6.8–11.8)	0.036
	*p*-value	0.002	<0.001	
School type (high school only)	General	3.0 (1.7–4.4)	9.0 (7.0–11.0)	<0.001
	Specialized ^2^	3.4 (1.9–4.8)	8.7 (4.9–12.5)	0.002
	*p*-value	0.722	0.902	
**Community characteristics**				
Community type	Urban (Dong)	9.0 (7.4–10.7)	16.6 (14.5–18.6)	<0.001
	Rural (Eup and Myeon)	10.8 (6.1–15.4)	16.5 (8.6–24.5)	0.186
	*p*-value	0.469	0.996	
City type	Large cities ^3^	10.1 (7.9–12.3)	17.0 (13.9–20.1)	<0.001
	Provincial areas ^4^	8.4 (6.2–10.6)	16.2 (13.6–18.9)	<0.001
	*p*-value	0.298	0.714	
Seoul Capital Area ^5^	Yes	8.5 (6.0–11.1)	16.9 (14.0–19.8)	<0.001
	No	9.8 (8.0–11.6)	16.2 (13.4–19.0)	<0.001
	*p*-value	0.434	0.743	

Chi-squared tests were used to examine statistical difference. * *p*-values: *p*-values for between dataset differences (2006–2008 and 2014–2016); *p*-values of <0.05 indicate statistical significance. ^1^ With friends/alone: living in a school dormitory or boarding house. ^2^ Specialized high school: high schools that focus on cultivating talent in specific fields, providing practice-oriented education designed for students who wish to get a job after graduation rather than apply for college. ^3^ Large cities: metropolitan cities (Seoul, Busan, Daegu, Incheon, Daejeon, Gwangju, and Ulsan) and the Special Self-Governing City, Sejong. ^4^ Provincial areas: provinces (Gangwon, Gyeonggi, Chungbuk, Chungnam, Jeonbuk, Jeonnam, Gyeongbuk, and Gyeongnam) and the Special Self-Governing Province, Jeju. ^5^ Seoul Capital Area: areas in and around the capital city (Seoul), including Incheon and Gyeonggi.

**Table 4 children-10-00613-t004:** Association between early sexual debut and various socio-environmental characteristics, adjusted for grade and survey year.

Variables		Subgroups	Odds Ratio (95% CI)	
			Boys	Girls
**Family characteristics**				
Household income	Highest		1.70 (0.99–2.88)	1.77 (0.63–4.96)
	High		1.04 (0.63–1.73)	1.78 (0.73–4.32)
	Moderate		1.59 (1.06–2.38)	1.52 (0.60–3.81)
	Low		2.38 (1.20–4.71)	3.37 (1.12–10.11)
	Lowest		0.66 (0.30–1.56)	26.38 (6.64–104.80)
Residence type	With family		1.77 (1.33–2.36)	1.73 (0.98–3.03)
	With relatives		1.00 (0.36–2.81)	16.47 (2.92–92.93)
	With friends/alone ^1^		1.42 (0.55–3.65)	10.18 (1.75–59.19)
	Care facility		2.86 (0.80–10.21)	0.99 (0.13–7.77)
Paternal education	≥College		1.40 (0.97–2.01)	1.52 (0.64–3.60)
	Completed high school		1.96 (1.16–3.32)	2.34 (0.95–5.78)
	≤Junior high school		2.36 (1.02–5.42)	10.41 (2.73–39.72)
	Unknown/deceased		0.74 (0.41–1.34)	4.56 (1.94–10.72)
Maternal education	≥College		1.07 (0.70–1.64)	0.86 (0.39–1.91)
	Completed high school		2.76 (1.81–4.22)	2.15 (0.85–5.44)
	≤Junior high school		2.15 (0.87–5.34)	1.45 (0.17–12.46)
	Unknown/deceased		0.75 (0.42–1.31)	8.16 (3.46–19.24)
**School characteristics**				
Academic performance	Good~excellent		1.12 (0.76–1.65)	2.71 (1.35–5.43)
	Fair		1.91 (1.16–3.16)	1.83 (0.67–5.02)
	Poor~unsatisfactory		1.77 (1.11–2.83)	3.36 (1.59–7.10)
Composition	Coed		1.38 (0.99–1.92)	2.69 (1.62–4.47)
	Boys only		1.61 (1.05–2.47)	-
	Girls only		-	4.24 (1.59–11.33)
School type (high school only)	General		1.54 (1.06–2.22)	4.98 (2.03–12.23)
	Specialized ^2^		1.08 (0.59–1.97)	3.62 (1.27–10.33)
**Community characteristics**				
Community type	Urban (Dong)		1.47 (1.11–1.94)	2.61 (1.58–4.32)
	Rural (Eup and Myeon)		1.60 (0.76–3.38)	5.51 (1.83–16.59)
City type	Large cities ^3^		2.05 (1.41–2.99)	1.88 (0.92–3.84)
	Provincial areas ^4^		1.17 (0.81–1.69)	3.57 (1.98–6.46)
Seoul Capital Area ^5^	Yes		1.82 (1.18–2.82)	2.90 (1.30–6.45)
	No		1.20 (0.88–1.64)	2.62 (1.54–4.44)

Multiple logistic regressions were used to explore associations. A multiple logistic model was designed for each subgroup of socio-environmental characteristics, and 2006–2008 pooled data were used as a reference. Odds ratio of the 2014–2016 pooled data for each subgroup of socio-environmental characteristic variable was estimated. ^1^ With friends/alone: living in a school dormitory or boarding house. ^2^ Specialized high school: high schools that focus on cultivating talent in specific fields, providing practice-oriented education designed for students who wish to get a job after graduation rather than apply for college. ^3^ Large cities: metropolitan cities (Seoul, Busan, Daegu, Incheon, Daejeon, Gwangju, and Ulsan) and the Special Self-Governing City, Sejong. ^4^ Provincial areas: provinces (Gangwon, Gyeonggi, Chungbuk, Chungnam, Jeonbuk, Jeonnam, Gyeongbuk, and Gyeongnam) and the Special Self-Governing Province, Jeju. ^5^ Seoul Capital Area: areas in and around the capital city (Seoul), including Incheon and Gyeonggi.

## Data Availability

The KYRBS data are available from the Korea Disease Control and Prevention Agency (URL: https://www.kdca.go.kr/yhs/ (accessed on 18 August 2021)).

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
