# Peer review of "Trends in Early Sexual Initiation and Its Association with Socio-Environmental Factors among Korean Adolescents"

_children, 2023, doi:10.3390/children10040613_

Round 1

Reviewer 1 Report

This is a quite interesting and well written article focusing upon trends in early sexual initiation with socio-environmental factors among Korean adolescents.

All the parts of the article are well described, Introduction, materials and methods, statistical analysis with SPSS version 28.0 IBM, results, discussion and conclusion, supported by 4 Tables and by a supplementary Table 1. 

The 20 references are relevant to the text of the article. 

Reviewing this article, one can understand how significant is the sexual  education in adolescents and how important is the appropriate  sexual behavior for the physical and mental health of the Korean young boys and girls.    

The indifference to adolescent sexual behavior leads to an early involvement in sexual activity of the young Korean boys and girls. This was shown clearly in the trends about sexual behavior and early activity in the students of the first reference group (pooled data from yrs 2006-2008) and of the research group (pooled data from yrs 2014-2016 ), where the early sexual initiation is increased and is associated with certain socio-environmental factors, such as family, school, and community characteristics.

It is remarkable that students who live in care facilities and those who live with their relatives showed a high weighted percentage of reporting early sexual initiation the 2014-2016 pooled data. 

Therefore, socio-environmental considerations regarding early sexual initiation trends should be administered, focusing on the physical and social development of adolescents based on healthy and safe sexual life.

Finally, this article suggests the necessity of mandatory sex education in order to provide knowledge about healthy sexual life.

Author Response

Thank you for your compliments and summary. It is really helpful. 

Reviewer 2 Report

You should explain better the concep of sexuality as it is focused not as a cultural construction and you avoid to explain also the influece of religion on the expreson of adolescents sexuality.

What do you consider "early sexual initiation"? What are you comparing with? Socio economcal factors as well as climate factors have a deep influence in the expresion of sexualidad from 8 years old even in some Pacific countries or even in hot climate countries as Canary Islands in Span. Why early? You have to explain better this key concept. Sexuality is acompany as since we are born, do you really mean "intercourse" and "heterosexuality" as hegemonic concepts. You need to read Leonore Tiefer or some antropology researcher as Malinowski, even Freud or freudomarxism.

Your explanation about sexual education need more details and information. Obligatory sexual education do not avoid sexual behaviour, information is not linked with more experiences as high sociao class do not means represion.

Author Response

1) I really appreciate your comment about the influence of religion and climate on adolescents' sexual activity. However, the KYRBS does not collect such information. Therefore, it was difficult to explain its association with early sexual initiation in this study. I've revised the limitation section in consideration of your comments. Please check. 

2) Early sexual initiation was defined as experiencing sexual intercourse at 13 years and younger in this study based on previous research (doi: 10.1016/S1054-139X(99)000041-5 and doi: 10.1001/jamapediatrics.2019.0458). I believe timing of sexual debut is important and should be studied more because (i) adolescents are in an immature state of physical and mental development. they can experience disrupted and distorted physical, mental, and social functioning when they initiate sexual activity without having necessary knowledge and (ii) it is influenced by personal and social circumstances. such experience may affect their sexual behavior and overall health for the rest of their lives. I've revised the introduction section considering your comment. Please check. Thank you. 

3) Thank you for your comments regarding sexual orientation and thank you for suggesting important literature regarding this issue as well. I will make sure to read them. First off, I do not believe that intercourse and heterosexuality as hegemonic concepts. I believe one should be free to express his/her own sexuality without any prejudice or social barriers. In this study, however, it was difficult to consider "sexual orientation" due to following reasons: (i) the KYRBS does not specifically collect data regarding one's sexual orientation (ii) the reason why I focused on heterosexual intercourse among adolescents was that it could lead to unintended pregnancy as mentioned in the introduction section. 

4) Thank you for your comments regarding sexual education. I’ve added more information and details now in the revised version. Please check. 

Reviewer 3 Report

Reviewer Comment

Although the subject of the study is interesting and the subject is unique, there are many places in the manuscript that do not comply with the article writing system. Individual information should be added to the method section of the summary. The sources used in the introduction were very old. If an up-to-date study is planned, the sources used here should be selected from studies published in recent years as much as possible. These old sources that you use should be replaced with updates. The method section should be edited as a template. Method titles and title descriptions should be explained more systematically. The study design and participants section should be added and the flow chart of the individuals should be added. In the conclusion section, each table explanation should be made separately before the table. The discussion section was the weakest part of the manuscript. Data should be discussed with many more literature sources. A correct relationship could not be established between the results and the literature, and the interpretations at the end of the association should be away from personal interpretations. All these regulations are very important for the manuscript. Please make all the arrangements carefully.

Revision

1.      Information on the age range of the participants and the number of participants should be added to the method section of the summary.

2.      Page 2, line 45-47;Therefore, an examination of the age of first sexual intercourse with the opposite sex, which could lead to unintended teenage pregnancy, should be considered.” Add reference at the end of the sentence.

3.      Page 2, line 63-65; “In Korea, studies on sexual activity have not attracted attention because of their conservative cultural backgrounds. Specifically, research on adolescent sexual activity is scarce.” Include the sources of these studies that are more than one in each sentence.

4.      The sources you used in the introduction are out of date. Research sources are very old in years. If you are working on current developments, the sources you use should be as up-to-date as possible. Edit these resources in the introduction.

5.      Add the research hypotheses at the end of the introduction.

6.      6. Edit the general headings of the Material Method section. The method section should be written in a specific layout and template. 1. Study design, 2. Participants, 3. Measurement methods, 4. Data analysis. In this way, chapter headings should be arranged more systematically. The study plan should be explained in the study design section. The place of the study, the date of the ethics committee and the permission information should be given. In the participants section, information about the characteristics of the individuals should be given. Inclusion and exclusion criteria should be given here. In addition, the flow chart of the individuals should be added to this section. Information on what is evaluated in measurement methods should be given. More information should be given about the “Korea Youth Risk Behavior Web-based Survey (KYRBS)” survey. Before the content, what is this survey, who created it, what do the results mean, is there any validity and reliability?, how many questions does it consist of, information about them should be given. In the data analysis section, information on how the normality test of the data is performed and whether the data is parametric or non-parametric and accordingly which statistical analysis method is chosen should be added.

7.      In the Result section, it is seen that all the data are given and then the tables are given one after the other. This way of presentation is far from the systematic of article writing and is not suitable for tracking table information. For each table, the explanation text of the table should be given separately for each table.

8.      Page 4, line 159, 168, 177; In these sentences, statistical significance values were mentioned, but the significance value of p was not written. The significance value of p should be added for each sentence.

9.      At the bottom of each table, the statistical method used in the analysis of the data and the significance values of p should be added.

10.  Discussion section is far from general article writing system. It is seen that there is no direct relationship between the results of your research and the literature. While making the discussion, a relationship should be established between the research results and the literature, and as a result of this relationship, a comment should be made by avoiding personal opinions based on these data. There is no unity in your discussion. The links are broken. Some results are not correctly correlated with literature information. In addition, all results should be discussed in more depth with much more literature sources. You do not need to mention statistical significance expressions in the discussion section. This form of expression should be given in the result section. The main conclusion you found in the discussion should be given. Towards the end of the discussion, explanations are made without citing any sources. All this writing style is not a suitable method for writing a discussion. In line with the suggestions, organize the whole discussion with more literature information.

Author Response

1) The total number of participants is described in the method section in the revised version. Students aged 13 to 18 years were included in the analyses as mentioned in the method section. Please check once more. Thank you.

2) A reference is now added in the revised version. 

3) Two reference is now added in the revised version. 

4) I’ve added some more current literature as reference.

5) Study hypothesis is added in the revised version of the manuscript.

6) Headings and contents are revised as commented. Thank you for your detailed explanations. It was really helpful. 

Ethical statement is given at the end of the manuscript (right before the references section).

The selection process of study subjects included in the analysis is now given as a flow chart (Figure 1). Please check.

7) In the revised version, the data explanations are given before each table.

8) In the revised version, p values are added for each sentence. 

9) In the revised version of the manuscript, statistical method used in the analysis and significance values of p are added at the bottom of Table 2~4.

10) Thank you for your comment on the discussion section. I've revised the discussion and reorganized some parts. Please check if it is now appropriate. 

Reviewer 4 Report

Thank you for the opportunity to revise this manuscript.

The paper deals with an interesting, timely and overlooked issue. The practical implications are also relevant. However a significant work of revision is needed in order to ameliorate and valorize these interesting aspects.

Line 23: more keywords should be added, like: “sexual intercourse”; “early sexual initiation”

Line 26-29: this paragraph should be rephrased and expanded

In the introduction a definition of “early sexual initiation” should be provided. In the introduction the author seems to use the term “early sexual initiation” and “sexual activity in adolescence” as synonyms. However, in the abstract a different definition is given.

Line 60: “However, only a few researchers have studied trends in early sexual initiation associated with socio-environmental factors”. The author should briefly describe the results from this study, in order to provide the literature context of his study

Line 63-64: the author should provide a reference for this, instead it seems like a personal opinion.

Line 63-68: the author should better expand this paragraph, explaining the aims of the study. For instance, in lines 66-67 he states: “to confirm whether there is a notable change in the age of sexual initiation”: what does he intends to say? Is this sentence referred to some previous study? If this is the case, adequate reference and explanation is needed.

The author could also underline (and provide adequate contextual theoretical framework) about the opportunity to take into consideration the role of gender, as in the analysis this aspect was taken into account (see for instance doi: 10.1016/j.esxm.2020.03.003  ; doi: 10.1016/j.esxm.2021.100405 )

Line 83-87: “From 2006 to 2008, 221,340 students completed the 83 surveys, and 10,255 students (4.6%) reported sexual intercourse with the opposite sex. The 84 total number of students who reported sexual intercourse at the time of the surveys (2014–85 2016) was 8,682, and the total number of survey participants in this period was 205,631 86 (4.2%).” This paragraph should be shifted in the Results section. Moreover, it should be rephrased. “at the time of the surveys” means: “in the second survey”?

The author should better explain why he decided to focus only on heterosexual early sexual initiation and also underline this aspect in the limitation section.

Line 120-129: the content of this paragraph seems in contrast with the abstract, where the author states that early sexual initiation is defined as having sexual intercourse at the age of 13 or before.

Line 226: “it was confirmed”: why the author say “confirmed”? It was not within the hypothesis. Is it a confirmation of some previous study? If this is the case, the author should specify it and report the previous data

Line 242-255: I find this paragraph out of topic, in consideration of the aim of the paper. Indeed, the paper is not a critique to the KYRBS survey.

Line 256-258: according to results, the change is more prominent for girls than for boys, however the percentage remains still higher for boys (19%) than for girls (16%). This aspect should be reported and adequately commented, providing adequate references and discussion. According to the literature, indeed, boys tend to be more prone to sexual risk behaviors, and risk behaviors in general, than girls (see for instance doi: 10.1016/j.esxm.2020.03.003)

In the analysis section, I think it could be interesting to consider also gender (or sex assigned at birth) as an independent variable in influencing different rates of early sexual initiation.

Finally, in the conclusion section a better focus on the practical implications of the results of the study could give more sterngth to the publication.

Author Response

1) I used the MeSH Browser (https://meshb.nlm.nih.gov/search) for choosing the keywords. The entry term "sexual intercourse" referred me to "coitus" which is already included as a keyword in the manuscript. Entry terms such as 'coital frequency'. 'first intercourse', 'sexual intercourse' all refer to 'coitus' automatically. There were no results for 'early sexual initiation' or 'sexual initiation.'

3) The study is now briefly added in the revised version. 

4) Appropriate reference is now added. Please check. 

5) A hypothetical statement is now added before the sentences. Please check. Thank you.

7) The statements are now shifted to the method section according to your and Reviewer 3's comments (I believe you meant the method section?). 

Since the KYRBS is conducted annually, "at the time of the surveys" meant "at the time the participants completed the survey"

In addition, the KYRBS does not collect specific data regarding sexual orientation so that only heterosexual intercourse was analyzed and it is now mentioned in the discussion section as a limitation. 

2,8) The statement in the abstract is now moved to the method section. There is no consensus regarding the timing of early sexual initiation since cultural and social backgrounds affect the initiation so that every country has different situation. A study conducted in 2003 defined early sexual initiation as 14 and younger. In this study, early sexual initiation was defined operatively based on other current previous research ( doi: 10.1016/S1054-139X(99)000041-5 and doi: 10.1001/jamapediatrics.2019.0458)

9) In the introduction section, a hypothetical statement is now added according to your and Reviewer 3’s comments. 

10) Thank you for your comment. As you mentioned, the purpose of this study is not to criticize the KYRBS. However, I believe that having an adequate monitoring system for adolescent sexual behavior is really important regarding exploring changes in the sexual activity patterns among adolescents. As mentioned in the manuscript, the KYRBS is the one and only monitoring system of Korean adolescents’ sexual behavior, but its structure is weak and not adequately managed. Scientific evidence regarding sexual behavior of adolescents is based on such monitoring system so that I hope you could understand the necessity of this paragraph and let it remain as it is. Thank you in advance.

11) The following statement is now added in the revised version: “However, caution should be taken when interpreting the result because the percentage remains still higher for boys (19.0%) than for girls (16.6%). In addition, boys tend to be more prone to risky health behavior than girls in general.

6, 12) Thank you for your suggestion. However, considering gender as well as sex assigned at birth in this study was not available due to data limitation. The KYRBS does not collect data regarding one’s sexual orientation. Examining the association of early sexual initiation and gender could be another study topic. Thank you for your comment though.

13) The conclusion is revised in consideration of your comment. Please check. Thank  you. 

Round 2

Reviewer 3 Report

Reviewer Comment

In general, although many of the proposed revisions appear to have been made in the first manuscript, some chapters still require editing. In the method section of the summary, information on the number of persons and gender should be given. In addition, numerical data should be presented statistically at the conclusion of the Summary. Although there are deficiencies in the references in some parts of the introduction, the most important deficiency is that the sources used are old. These sources must be replaced with current ones. The regulations of the manuscript are given below. Please make all arrangements completely and carefully. With these regulations, the manuscript will reach a better level in quality.

Revision

1.      Information on the number of people in the study and their gender should be added to the method section of the summary.

2.      Numerical data containing the statistical results of the research should be added to the conclusion part of the abstract. In addition, p significance values should also be expressed.

3.      The first 12 articles you used as references in the introduction are very old. The articles you use should be much more up-to-date. Revise more.

4.      Page 2, line 59-63; “Some studies have targeted the protective factors for early sexual initiation. One of these studies, which was performed in the United States, found that living with parents, having a high socioeconomic level, living in rural areas, satisfactory academic achievement, and faithfulness were protective factors against postponing sexual activity during adolescence [5]” If you are talking about more than one research, you should include the sources of these studies at the end of the sentence. It is not possible to emphasize more than one work with a single reference. Edit this.

5.      Page 2, line 63-65; However, only a few researchers have studied trends in early sexual initiation associated with socio-environmental factors [16]. If you are using a statement such as several researchers, you should cite these studies as a source.

6.      Page 2, line 65-67;A study conducted in the United States in 2019 revealed that rates of early sexual initiation varied by race/ethnicity, location, and maternal educational level. Yet, such research has not yet been conducted in Korea. Please include the source of this study.

7.      Page 2, line 69-71; “Specifically, research on adolescent sexual activity is scarce.” Include references to the studies that you have mentioned as a small number.

8.      Page 3, line 111-113; The image quality of the texts in Figure 1 is very low. The texts cannot be read. Take higher quality image recording to make the text in Figure 1 more legible.

9.      Page 4, line 125 “2.2.2. Socio-environmental factors” KYRBS explanation has been made under this title, but the relevant source has not been added to the paragraph. Add reference at the end of the paragraph.

10.  Page 4, line 149; In the statistical analysis section, information on how the normality of the data is evaluated, whether the data are normally distributed, and whether parametric or non-parametric tests are used accordingly, should be added.

11.  Page 10, line 284-289; In the United States, health education is mandatory in most states and formally includes sex ed-ucation in the curriculum. However, the educational situation in Korea is different in that sex education is not included in the regular curriculum, and intermittent education is provided once a year by a lecturer or a health teacher, who is a licensed nurse positioned at school for care injured or ill students primarily. Include references after each study description, as these are studies when citing American and Korean studies.

Author Response

Once again, thank you for your comments and suggestions along with your time. I've revised the manuscript accordingly. 

1,2. Due to abstract word limits, it would not be possible to add more information to the abstract. I hope your understanding. Thank you in advance. 

3. I've deleted one reference from 1976 and replace it with one from 2008. The first two references ([1][2]) are quite old as you had mentioned. However, considering the contents of the following paragraph, I believe using reference from the twentieth century are necessary. I hope you would kindly understand my intention. Thank you in advance. 

4. More reference is now added. 

5. I've revised the statement now. Please check. 

6. I've mistakenly missed the reference. Reference is not included. Thank you for pointing this out. 

7. References are now added. 

8.  The resolution of the image is 400 DPI which is considered as pretty high. I've made the figure a bit more larger in the manuscript. I've uploaded the original figure image on the web server. I don't know whether you could check it or not. If possible, please check. 

9. Reference is now added. Please check. 

10. In this study, parametric tests were used based on the central limit theorem. Regression model fitness was check using AIC criteria. Such information is basic statistical analysis knowledge so that I think it would rather not describe in specific in the manuscript. I hope your understanding. Thank you in advance. 

11. Reference is now included for each statements. 

Reviewer 4 Report

Overall, the quality of the revised manuscript is improved. 

Note that the author's reply to reviewers should be provided with a point-by-point answer, citing the single reviewer's comments. Yours reponse is confusing and unclear, as it is not clear to what comments yo give answers. Moreover, you have put together the comments of all reviewers.

One only further suggestion: 

Line 67-70: "Specifically, research on adolescent sexual activity 69 is scarce. In this study, it was hypothesized that more Korean adolescents would experience early sexual intercourse over time" The authors should contextualize their hypothesis. Based on what previous data from the literature do they hypothesize this? WIthout this information, it seems that the hypothesis is casual.

Author Response

Again, thank you so much for your thoughtful comments. 

Previous responses for all four reviewers can be found in the attached file. 

I am sorry about the inconvenience. 

The hypothetical statement is now revised according to your suggestion. 

Thank you again. 
